# Rationally Designed Minimal Bioactive Domains of AS-48 Bacteriocin Homologs Possess Potent Antileishmanial Properties

Hannah N. Corman,[a,b] Jessica N. Ross,[a,b] Francisco R. Fields,[c] Douglas A. Shoue,[a,b] Mary Ann McDowell,[a,b] Shaun W. Lee[a,b]

aUniversity of Notre Dame, Department of Biological Sciences, Notre Dame, Indiana, USA
bUniversity of Notre Dame, Eck Institute for Global Health, Notre Dame, Indiana, USA
cJohnson & Johnson Consumer Products Inc., Skillman, New Jersey, USA

Hannah N. Corman and Jessica N. Ross contributed equally to this article. Author order was determined in order of decreasing seniority.

**ABSTRACT** Leishmaniasis, a category I neglected tropical disease, is a group of diseases caused by the protozoan parasite *Leishmania* species with a wide range of clinical manifestations. Current treatment options can be highly toxic and expensive, with drug relapse and the emergence of resistance. Bacteriocins, antimicrobial peptides ribosomally produced by bacteria, are a relatively new avenue for potential antiprotozoal drugs. Particular interest has been focused on enterocin AS-48, with previously proven efficacy against protozoan species, including *Leishmania* spp. Sequential characterization of enterocin AS-48 has illustrated that antibacterial bioactivity is preserved in linearized, truncated forms; however, minimal domains of AS-48 bacteriocins have not yet been explored against protozoans. Using rational design techniques to improve membrane penetration activity, we designed peptide libraries using the minimal bioactive domain of AS-48 homologs. Stepwise changes to the charge ($z$), hydrophobicity ($H$), and hydrophobic dipole moment ($\mu H$) were achieved through lysine and tryptophan substitutions and the inversion of residues within the helical wheel, respectively. A total of 480 synthetic peptide variants were assessed for antileishmanial activity against *Leishmania donovani*. One hundred seventy-two peptide variants exhibited 50% inhibitory concentration ($IC_{50}$) values below 20 $\mu$M against axenic amastigotes, with 60 peptide variants in the nanomolar range. Nine peptide variants exhibited potent activity against intracellular amastigotes with observed $IC_{50}$ values of $<4$ $\mu$M and limited *in vitro* host cell toxicity, making them worthy of further drug development. Our work demonstrates that minimal bioactive domains of naturally existing bacteriocins can be synthetically engineered to increase membrane penetration against *Leishmania* spp. with minimal host cytotoxicity, holding the promise of novel, potent antileishmanial therapies.

**IMPORTANCE** Leishmaniasis is a neglected tropical disease caused by protozoan parasites of the genus *Leishmania*. There are three primary clinical forms, cutaneous, mucocutaneous, and visceral, with visceral leishmaniasis being fatal if left untreated. Current drug treatments are less than ideal, especially in resource-limited areas, due to the difficult administration and treatment regimens as well as the high cost and the emergence of drug resistance. Identifying potent antileishmanial agents is of the utmost importance. We utilized rational design techniques to synthesize enterocin AS-48 and AS-48-like bacteriocin-based peptides and screened these peptides against *L. donovani* using a fluorescence-based phenotypic assay. Our results suggest that bacteriocins, specifically these rationally designed AS-48-like peptides, are promising leads for further development as antileishmanial drugs.

**KEYWORDS** AS-48, *Leishmania*, antileishmanial, antimicrobial peptides, antiparasitic agents, bacteriocins

Address correspondence to Shaun W. Lee, Shaun.W.Lee.310@nd.edu.

The authors declare no conflict of interest.

[This article was published on 7 November 2022 with the names in the byline in a different order. The byline was updated in the current version, posted on 16 November 2022.]

Leishmaniasis is a vector-borne, parasitic disease caused by protozoan parasites of the *Leishmania* genus (1, 2). The disease is endemic in 98 countries, placing almost 300 million people at risk of infection each year (3) and resulting in an estimated 2.4 million disability-adjusted life years (DALYs) (4). Leishmaniasis has a wide spectrum of clinical manifestations, from self-healing skin lesions to hepatosplenomegaly and fatality. There are three primary clinical forms: visceral, cutaneous, and mucocutaneous leishmaniasis (5). Visceral leishmaniasis, the deadliest form, results in a 95% mortality rate if left untreated (6). The World Health Organization (WHO) has declared leishmaniasis a category I neglected tropical disease, and as such, it does not receive adequate attention related to its burden (7).

There are limited viable treatment options currently available for leishmaniasis (8). In resource-limited areas where leishmaniasis is prevalent, pentavalent antimonials are prescribed for all manifestations; however, these compounds are available only intravenously, exhibit high toxicity, and require prolonged treatment, thus resulting in increased clinical and financial burdens (9–11). Furthermore, increasing numbers of cases of drug resistance to pentavalent antimonials have been observed (12, 13). Due to incidents of disease relapse as a result of drug resistance, other agents such as amphotericin B, paromomycin, and miltefosine are being used (14). Unfortunately, these treatment options are costly and have high cellular toxicity, and drug resistance has also been observed (15). There is a great need for new, safe, and effective antileishmanial therapeutics.

Antimicrobial peptides (AMPs) are an intriguing avenue as alternative antiprotozoal agents or to complement current antiprotozoals (16, 17). AMPs are highly diverse peptides, which span the range from prokaryotes to lower and higher eukaryotes (18). Many AMPs are membrane active and have been previously shown to have potent antiparasitic activity (19, 20). Specifically, natural and synthetic cationic AMPs have been shown to be effective against protozoan parasites such as *Trypanosoma cruzi* (21), *Plasmodium* (22), and *Leishmania* (23). Leishmanicidal activity has been observed with histatin 5, a human salivary AMP, which gains entry to cells via membrane targeting, causing the consequential depolarization of the membrane, and additionally targets mitochondrial ATP synthesis in *Leishmania* spp. (24). Bombinins H2 and H4, isolated and purified from the skin secretions of the frog species *Bombina variegata*, have also demonstrated potent antileishmanial activity (25, 26). Due to their proven efficacy, high specificity, decreased drug interactions, and low toxicity, AMPs may be untapped sources of novel antileishmanial drug therapies.

Of particular interest in the exploration of novel antimicrobial drugs are bacteriocins, which are AMPs produced by bacteria (27). These ribosomally produced peptides have been shown to have bactericidal (28, 29), fungicidal (30, 31), and parasiticidal (32) properties, giving promise as an alternative to conventional chemical-based antimicrobials. Bacteriocins are characterized into three classes based on size, content, stability, and posttranslational modification (33). Class I (modified) and class II (unmodified) bacteriocins both start as a propeptide containing a leader sequence, which is then cleaved during ribosomal translation, resulting in the functional peptide (34). Posttranslationally, class I has further subgroup classifications based on specific modifications, such as heterocyclization, glycosylation, and head-to-tail circularization. Class III bacteriocins are large molecules of >10 kDa and are thermolabile. Nisin, a class I lantibiotic produced by *Lactococcus lactis*, has been FDA approved for use as a food preservative (35). Another lantibiotic, GP15cin, is used in ethanol fermentation plants to control inappropriate bacterial growth (36). Bacteriocins are a promising alternative to clinical antimicrobials, as antimicrobial resistance is increasing.

Due to their proteinaceous nature, bacteriocins can be modified via biochemical engineering techniques to investigate bioactive domains and improve specific functions (37). Biochemical engineering techniques to modify AMPs for antimicrobial therapies have been explored using site-directed mutagenesis, *de novo* design, and template-assisted approaches to create synthetic peptide libraries for evaluation (38–40). In order to discern residues that may play a key role in the antibacterial properties of AMPs, unbiased alanine substitutions can be made for every amino acid, building large

libraries to screen for antimicrobial activity (41). This technique creates a blueprint of which residues may negatively or positively impact bioactivity. Similarly, the intentional truncation of bioactive peptides may aid in the identification of bioactive regions to further understand areas of interest for future modification (42). The *de novo* design of bacteriocins can be employed to generate random or template-assisted peptides, which can subsequently be assessed computationally using biochemical and biophysical parameters (43). Template-assisted approaches evaluate naturally occurring AMPs to identify biochemically active domains, which can then be used as a template in a rational design approach to optimize bioactivity (44). Rational design is a computational approach that introduces minimal, stepwise changes to an amino acid sequence of a scaffold peptide to improve bioactivity.

Enterocin AS-48, a class I bacteriocin, has been well characterized and is currently being investigated for use as an antimicrobial drug against a large variety of infectious disease agents (45). Enterocin AS-48 is a ribosomally produced, circularized bacteriocin encoded by *Enterococcus faecalis* subsp. *liquefaciens* strain S-48, which undergoes head-to-tail macrocyclization as a posttranslational modification (46). Efforts to engineer synthetic enterocin AS-48 have been made in order to improve mass production and bioactivity. Previously, segments of the bacteriocin have undergone $\alpha$-ketoacid-hydroxylamine ligation to chemically produce synthetic enterocin AS-48 in a cyclical form (47). However, due to the ligation process, its bioactivity decreased compared to that of the natural product. Furthermore, using limited proteolysis, linear versions of enterocin AS-48 have been purified (48). Truncation of the linear form identified that the membrane-penetrating activity of enterocin AS-48 is attributed to a specific $\alpha$-helical region on the circular peptide, which largely contains cationic residues (49). Previously, we utilized specific rational design techniques using AS-48 bacteriocin homologs to design a series of reductive peptide variants of AS-48 optimized for membrane-penetrating bioactivity, which subsequently demonstrated potent antibacterial activity (50). Other studies have shown that full-length enterocin AS-48 induced mitochondrial damage to *Leishmania* promastigotes and also exhibited 50% inhibitory concentration ($IC_{50}$) values against *Leishmania pifanoi* axenic amastigotes of $10.2 \pm 1.2$ $\mu$M (51). Due to this previously confirmed parasiticidal activity, we hypothesized that AS-48-based minimal domain peptides can also exhibit antileishmanial properties. In this paper, we demonstrate the use of rationally designed minimal AS-48 bacteriocin homologs for antileishmanial drug candidate identification; 9 of the 480 peptides exhibited potent activity against *Leishmania donovani* intracellular amastigotes with $IC_{50}$ values of less than 4 $\mu$M, which is within the target product profile (TPP) established by the Drugs for Neglected Diseases Initiative (DNDi) (50).

## RESULTS

**Peptide library screen and axenic amastigote $IC_{50}$ determination.** We utilized a previously developed high-throughput screening assay using transgenic axenic amastigotes expressing a red fluorescent protein, mCherry. Changes in mCherry fluorescence were normalized to 50 $\mu$M miltefosine in order to quantify parasite inhibition (52). In total, 5 scaffold 25-mer peptides and 475 synthetic 25-mer peptide variants were screened (Fig. 1). Due to previous data showing that enterocin AS-48 exhibited an $IC_{50}$ value of approximately 20 $\mu$M, each peptide was screened at 20 $\mu$M to identify minimal AS-48 bacteriocin peptide variants that outperform enterocin AS-48. The assay was repeated, and the average percent parasite killing was determined. A total of 172 peptide variants killed at least 80% of the parasites and were selected for further study. Of the 172 peptide variants that exhibited potent antileishmanial activity, 9 were syn-enterocin, 32 were syn-larvacin, 42 were syn-safencin, 32 were syn-sordellicin, and 57 were syn-xiamencin. Axenic amastigote $IC_{50}$ value determination revealed that 137 variants had $IC_{50}$ values of <5 $\mu$M, while 60 of those peptide variants had $IC_{50}$ values in the nanomolar range (see Tables S2 to S6 in the supplemental material).

**Cytotoxicity against THP-1 macrophages.** The 172 peptide variants with antileishmanial activity were then screened against THP-1 macrophages to determine host cell

**FIG 1** Experimental workflow to determine potent leishmanicidal peptides. Four hundred eighty peptide variants were screened at 20 $\mu$M and advanced to the next round if 80% effective compared to the miltefosine control. THP-1 macrophages were used to establish $CC_{50}$ values, subsequently advancing peptides to the next round if the values were above 20 $\mu$M. Ten peptide variants found to have an SI value of >20 and an $IC_{50}$ value of <10 $\mu$M against axenic amastigotes were tested against intracellular amastigotes.

toxicity. The peptides were screened at 20 $\mu$M to identify peptide hits, which exhibited $IC_{50}$ values below the previously characterized enterocin AS-48 $IC_{50}$ (51). Twenty-five peptide variants exhibited limited host cell toxicity, with 50% cytotoxic concentration ($CC_{50}$) values above 20 $\mu$M and selectivity index (SI) values ranging from 1.40 to 74.07 (Table 1). Of these 25 variants, 4 were syn-enterocin, 3 were syn-larvacin, 10 were syn-safencin, 3 were syn-sordellicin, and 5 were syn-xiamencin. Ten peptide variants, which exhibited $CC_{50}$ values above 20 $\mu$M, exhibited axenic amastigote $IC_{50}$ values in the nanomolar range and consequently had an SI of >20. Of these peptide variants, three were syn-enterocin, one was syn-larvacin, and six were syn-safencin. Subsequentially, these 10 peptides were characterized as potential antileishmanial drug candidates and were evaluated further for future clinical use.

**Intracellular amastigote $IC_{50}$ determination.** As *Leishmania* spp. are obligate intracellular parasites, intracellular amastigote $IC_{50}$ value determination is necessary to understand the clinical relevance of the 10 antileishmanial peptide drug candidates. Initially, peptides were screened against intracellular amastigotes at 20 $\mu$M to confirm efficacy, as enterocin AS-48 showed lethality only against promastigotes and axenic amastigotes, with limited activity against intracellular amastigotes. All 10 peptides were observed to have $IC_{50}$ values below 20 $\mu$M; thus, $IC_{50}$ values were established against intracellular amastigotes. Nine of the peptides exhibited $IC_{50}$ values at or below 4 $\mu$M, including syn-safencin 82, which exhibited potent activity against the intracellular amastigotes, as revealed by an $IC_{50}$ value of 1.0 $\mu$M (Fig. 2 and Table 2). Syn-larvacin 35 had an intracellular amastigote $IC_{50}$ value approximately 38 times higher than the initial axenic amastigote $IC_{50}$ value, likely indicating that the peptide was unsuccessful in crossing the macrophage barrier to reach the intracellular parasite.

**TABLE 1** Cytotoxicity screening against THP-1 cells with corresponding axenic amastigote IC$_{50}$ values and consequential selectivity index values[a]

| Peptide variant | THP-1 macrophage CC$_{50}$ ($\mu$M) | Axenic amastigotes | |
| --- | --- | --- | --- |
| | | Mean IC$_{50}$ ($\mu$M) ± SD | SI |
| Syn-enterocin 39 | >20 | 0.27 ± 0.05 | 74.07 |
| Syn-enterocin 47 | >20 | 0.28 ± 0.07 | 71.68 |
| Syn-safencin 73 | >20 | 0.35 ± 0.08 | 56.50 |
| Syn-enterocin 15 | >20 | 0.36 ± 0.03 | 55.71 |
| Syn-safencin 82 | >20 | 0.375 ± 0.10 | 53.33 |
| Syn-safencin 77 | >20 | 0.39 ± 0.13 | 51.28 |
| Syn-safencin 6 | >20 | 0.48 ± 0.07 | 41.67 |
| Syn-larvacin 35 | >20 | 0.63 ± 0.14 | 31.95 |
| Syn-safencin 7 | >20 | 0.77 ± 0.08 | 25.91 |
| Syn-safencin 78 | >20 | 0.94 ± 0.14 | 21.19 |
| Syn-enterocin 69 | >20 | 1.03 ± 0.13 | 10.95 |
| Syn-safencin 94 | >20 | 1.17 ± 0.07 | 17.05 |
| Syn-xiamencin 3 | >20 | 1.26 ± 0.17 | 15.87 |
| Syn-safencin 2 | >20 | 2.42 ± 0.09 | 8.26 |
| Syn-xiamencin 41 | >20 | 2.73 ± 0.15 | 7.33 |
| Syn-xiamencin 43 | >20 | 3.54 ± 0.08 | 5.65 |
| Syn-safencin 66 | >20 | 3.64 ± 0.10 | 5.49 |
| Syn-larvacin 14 | >20 | 3.89 ± 0.10 | 5.14 |
| Syn-larvacin 69 | >20 | 4.14 ± 0.11 | 4.84 |
| Syn-xiamencin 52 | >20 | 5.23 ± 0.06 | 3.82 |
| Syn-xiamencin 42 | >20 | 6.83 ± 0.09 | 2.93 |
| Syn-sordellicin 36 | >20 | 7.59 ± 0.07 | 2.63 |
| Syn-safencin 72 | >20 | 12.07 ± 0.09 | 1.66 |
| Syn-sordellicin 19 | >20 | 12.98 ± 0.13 | 1.54 |
| Syn-sordellicin 6 | >20 | 14.25 ± 0.17 | 1.40 |

[a]Values listed are the means from three replicates.

**Secondary structure and thermostability analyses.** Computational analysis of secondary structures revealed that all 10 antileishmanial drug candidates took an $\alpha$-helical structure (Fig. 3). Syn-safencin 73 had a decreased $\alpha$-helical structure, taking the conformation in the fragment window of $_3$KETIRQYLKNEIKKK$_{17}$. Helical wheel predictions along with hydrophobic moment ($\mu$H) quantification revealed that amphipathicity may not be a favorable biochemical characteristic in antileishmanial peptide design. Within the peptide libraries, peptide variants with increased amphipathicity, achieved through the inversion of residues within the helical wheel and confirmed using HeliQuest, were not initial hits, indicating an axenic amastigote IC$_{50}$ value of >20 $\mu$M. Syn-larvacin 35, a clear outlier after the determination of the intracellular amasti-

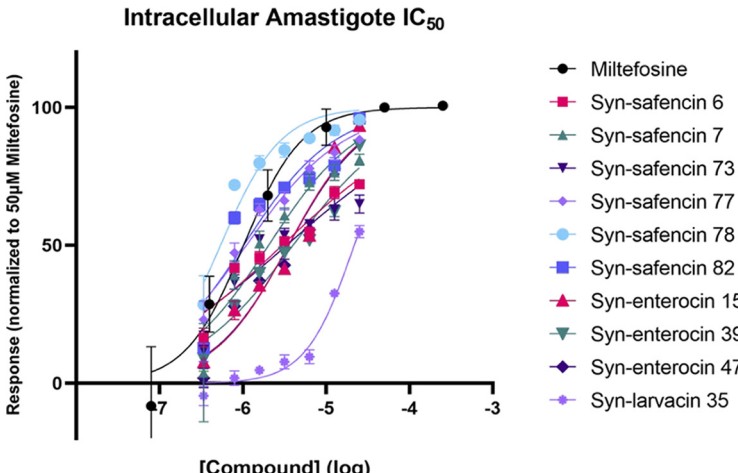

**FIG 2** Intracellular amastigote IC$_{50}$ curves of 10 antileishmanial drug candidates. Values listed are the means from three replicates.

**TABLE 2** Intracellular amastigote $IC_{50}$ and selectivity index values[a]

| Peptide variant | Mean $IC_{50}$ ($\mu$M) ± SD | SI |
|---|---|---|
| Syn-enterocin 39 | 4.00 ± 0.04 | 5.00 |
| Syn-enterocin 47 | 3.54 ± 0.04 | 5.66 |
| Syn-safencin 73 | 3.35 ± 0.08 | 5.97 |
| Syn-enterocin 15 | 3.52 ± 0.04 | 5.69 |
| Syn-safencin 82 | 1.02 ± 0.07 | 19.55 |
| Syn-safencin 77 | 1.59 ± 0.04 | 12.59 |
| Syn-safencin 6 | 2.97 ± 0.05 | 6.74 |
| Syn-larvacin 35 | 23.67 ± 0.02 | 0.85 |
| Syn-safencin 7 | 2.04 ± 0.06 | 9.82 |
| Syn-safencin 78 | 3.89 ± 0.06 | 5.14 |

[a]Values listed are the means from three replicates.

gote $IC_{50}$ value, was observed to have an increased mean hydrophobicity (*H*) (*H* = 0.206) and a net charge (*z*) (*z* = 6) in the lowest range observed relative to the other peptide variant hits (Table 3). The most potent peptide with the highest SI value from the intracellular amastigote $IC_{50}$ determination assay, syn-safencin 82, was observed to have the highest net charge (*z* = 8). The majority of the antileishmanial peptide drug candidates were observed to have a net charge of 7, including the second most promising drug candidate, syn-safencin 77. Interestingly, syn-safencin 77 was also observed to have the lowest hydrophobicity, at 0.074.

Quantification of the melting temperature, used as an indicator of thermostability, revealed a large range of melting temperatures from 60.0°C to 102.1°C (Table 3). The most thermostable antileishmanial drug candidate was observed to be syn-safencin 73, with a melting temperature of 102.1°C. Further analysis of the melting temperatures of all axenic amastigote peptide hits showed the high thermostability of both syn-safencin and syn-sordellicin peptide variants, with many peptides having melting temperatures of >100°C; however, the syn-larvacin peptide variants had decreased thermal stability, as observed by melting temperatures of <60°C. The change in the Gibbs free energy (Δ*G*) was used as an

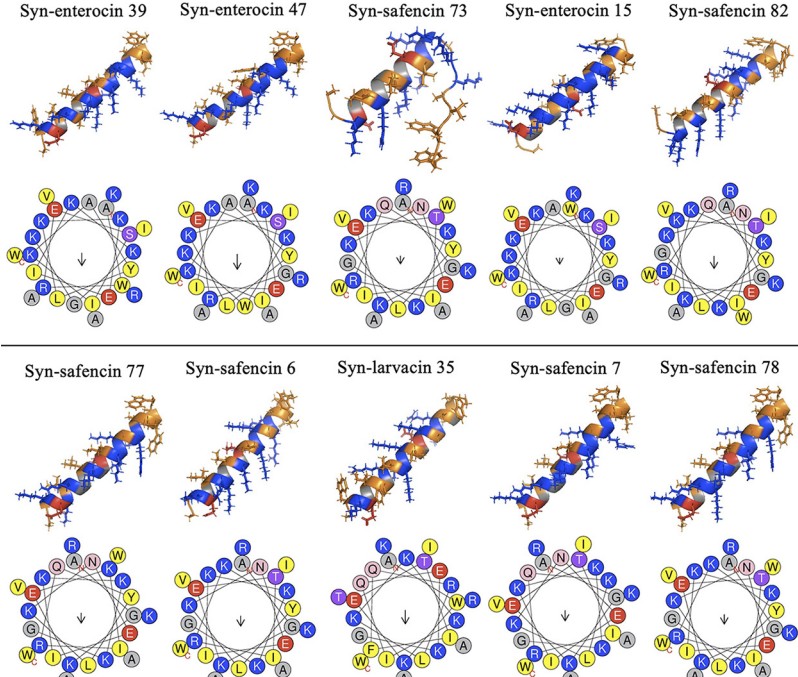

**FIG 3** PEPFOLD3 secondary structure predictions for antileishmanial drug candidates, visualized in PyMOL. Hydrophobic regions are in orange, positive regions are in blue, and negative regions are in red. The helical wheel was computationally predicted using HeliQuest. The arrow size indicates the relative hydrophobic moment.

**TABLE 3** Biochemical characterization and stability analysis of antileishmanial drug candidates

| Peptide variant | z | H | $\mu H$ | Melting temp (°C) | $\Delta G$ (kcal/mol) |
|---|---|---|---|---|---|
| Syn-enterocin 39 | 7 | 0.19 | 0.242 | 78.0 | −3.8 |
| Syn-enterocin 47 | 7 | 0.19 | 0.288 | 86.6 | −4.5 |
| Syn-safencin 73 | 6 | 0.124 | 0.151 | 102.1 | −4.5 |
| Syn-enterocin 15 | 7 | 0.178 | 0.123 | 77.0 | −2.5 |
| Syn-safencin 82 | 8 | 0.17 | 0.238 | 83.9 | −4 |
| Syn-safencin 77 | 7 | 0.074 | 0.191 | 70.3 | −3.8 |
| Syn-safencin 6 | 7 | 0.076 | 0.194 | 73.0 | −2.7 |
| Syn-larvacin 35 | 6 | 0.206 | 0.259 | 61.0 | −5.2 |
| Syn-safencin 7 | 7 | 0.028 | 0.196 | 60.0 | −2.1 |
| Syn-safencin 78 | 7 | 0.094 | 0.181 | 74.2 | −2.8 |

indication of thermodynamic stability, subtracting the relative energies of the folded state ($G_f$) from those of the unfolded state ($G_u$); thus, the more negative the value, the more thermodynamically favorable. Thermodynamic stability varied from −2.1 to −5.2 kcal/mol within the 10 antileishmanial drug candidates. Among all initial peptide hits, $\Delta G$ ranged from −0.4 to −10.6 kcal/mol, indicating large variations in thermodynamic stability.

## DISCUSSION

In this work, we determined the efficacy and cytotoxicity of 480 synthetic, minimal AS-48 bacteriocin homolog peptide variants against *L. donovani* using phenotypic in vitro fluorometric assays. These peptide variants have been shown to have potent antibacterial properties but have not been previously identified as antileishmanial agents. This study illustrates the successful use of rational design using the minimal domain of naturally occurring bacteriocin scaffolds to increase leishmanicidal activity. Specifically, enterocin AS-48 previously exhibited $IC_{50}$ values against axenic amastigotes of 10.2 to 19.5 $\mu$M depending on the *Leishmania* species (49). However, with rationally designed minimal AS-48 bacteriocin homologs, we identified 137 minimal peptide variants with $IC_{50}$ values against axenic amastigotes of <5 $\mu$M, 60 of which were in the nanomolar range. Importantly, these peptide variants had $IC_{50}$ values similar to those reported previously for AS-48 against Gram-positive bacteria (53–55) and lower $IC_{50}$ values than those of many membrane-active eukaryotic peptides (56, 57). Furthermore, we identified nine variants with $IC_{50}$ values at or below 4 $\mu$M against intracellular amastigotes, while enterocin AS-48 was previously shown to have limited activity against the intracellular parasite (49), a more clinically relevant analysis of drug efficacy in vitro. The nine antileishmanial peptides also exhibited limited host cell toxicity against THP-1 macrophages; thus, syn-enterocins 15, 39, and 47 and syn-safencins 6, 7, 73, 77, 78, and 82 are worthy of further characterization and development as potential chemotherapeutics.

The peptide variants that we assessed were rationally designed to increase the net charge (z), hydrophobicity (H), and hydrophobic moment ($\mu H$), a quantitative measurement of amphipathicity. These changes were achieved through the replacement of the short-chain amino acids alanine and glycine with lysine, the replacement of aliphatic and nonpolar short-chain amino acids with tryptophan, and the inversion of amino acid residues within the helical wheel. These 3 modifications are present in the nine potent peptide variants. When comparing the final three intracellular $IC_{50}$ syn-enterocin hits, syn-enterocins 15, 39, and 47, to the syn-enterocin scaffold sequence, each peptide variant had amino acids 17 and 18, glycine and lysine, flipped. In addition, each of these peptide variants had a single amino acid replaced with tryptophan; syn-enterocin 15 has a tryptophan at position 1, syn-enterocin 39 has a tryptophan at position 2, and syn-enterocin 47 has a tryptophan at position 17. When comparing the six syn-safencin peptide variants evaluated for intracellular $IC_{50}$ values to the syn-safencin scaffold sequence, a tryptophan substitution toward the C terminus was present in syn-safencins 73, 77, 78, and 82. In addition, five of these syn-safencin peptide variants

with potent intracellular activity had amino acids replaced with lysine toward the N terminus, between positions 4 and 9. The replacement of amino acids with tryptophan toward either the N terminus or the C terminus has been shown previously to improve antimicrobial activity (50), consistent with our findings that these key amino acid positions are critical for activity.

The replacement of short-chain amino acids with a positively charged lysine increases the cationic character of the peptide, and this is believed to increase the overall affinity of the peptide for the more anionic target membrane. Based on our observations, modifications toward either terminus of the peptide seem to have a substantial effect on peptide potency against *L. donovani*. Additionally, our most potent intracellular amastigote drug candidate, syn-safencin 82, had an increased net charge of 8. Comparatively, all other antileishmanial drug candidates exhibited a net charge of either 6 or 7. Interestingly, our computational analysis indicated that amphipathicity may not be an important biochemical property of optimized antileishmanial compounds. This was first illustrated when quantifying the amphipathicity of all peptide variants. Peptides with high amphipathicity did not exhibit antileishmanial properties during our initial screen against axenic amastigotes. Furthermore, all 10 final antileishmanial drug candidates had relatively low hydrophobic dipole moment values, supporting evidence that this biochemical parameter may not optimize membrane-penetrating activity against parasitic targets.

AS-48 homologs and their corresponding truncated, linearized forms are interesting potential chemotherapeutics for many infectious diseases. While we did not investigate the molecular mechanism of action of these peptide variants, the most frequently described model is through membrane disruption (58). The first step in this mechanism of action is initiated by electrostatic attraction between the cationic peptide and anionic phospholipids composing the membrane. Upon electrostatic interactions, peptides may dimerize (59, 60) within the membrane, forming pores (61), ultimately leading to cell death. The differences in the cell membrane lipid composition of the pathogen compared to human host cells is believed to be a key driver of the membrane recognition and selection processes (62). Host macrophages largely contain zwitterionic phospholipids throughout the outer leaflet, potentially preventing substantial electrostatic interactions with cationic peptides that would otherwise cause membrane pore formation and cellular cytotoxicity. This property would allow peptides to potentially be endocytosed in the host cell at concentrations sublytic to the macrophage but would allow the peptide to reach the intracellular parasite to exert activity (63). In addition, the more cationic peptides could preferentially interact with the anionic syndecans or heparan sulfate components found within the extracellular matrix of macrophages for endocytosis. *Leishmania* promastigotes have a surface glycolipid lipophosphoglycan (LPG) that plays critical pleiotropic roles in parasite survival and infectivity in both the sand fly and the mammalian host (64). Previous research has shown that enterocin AS-48 modified the fluidity of protozoal membranes and impaired the activity of several membrane-bound proteins, which provides some support for the potential membrane targeting of our AS-48-based peptide variants (51). However, *Leishmania* amastigotes have significantly less LPG, often corresponding to <100 molecules/cell (65, 66). While amastigotes express little to no LPG, research has shown that various parts of LPG exist as distinct entities, termed glycosylphosphatidylinositol (GPI) antigens (67, 68) or glycosylinositolphospholipids (GIPLs) (65) *L. donovani* amastigotes have been shown to synthesize GIPLs in quantities comparable to those reported previously for promastigotes (65). These GIPLs are less cationic than the highly cationic LPG, which could decrease the density of the peptides close to the plasma membrane. This information suggests that there could also be intracellular targets that we did not elucidate in this study; previous research proposed *Leishmania* mitochondria as likely targets due to a strong interaction between cardiolipin and AS-48 (51). An intracellular target could also explain the submicromolar efficacy, similar to the efficacy seen against other kinetoplastids like *Trypanosoma brucei* (32).

AMPs such as enterocin AS-48 have several characteristics that would be advantageous for further development as antileishmanial agents. Bacteriocins can be produced in large amounts at a limited cost (65), are highly stable with a compact

structure (66, 67), are resistant to exopeptidase degradation, and have low immunogenicity (68). Our study validates the use of rational design-based approaches to produce minimal domain peptide libraries that can be screened to identify lead peptide candidates with potent antiprotozoal activity and low host cell toxicity. Our library screen identified several minimal AS-48 bacteriocin-based peptides that retained strong leishmanicidal activity against clinically relevant intracellular amastigote forms. Our results prompt further studies to investigate minimal peptides as potent antileishmanial drug candidates, along with further screening of other protozoan parasites.

## MATERIALS AND METHODS

**Rational design and synthesis of AS-48-based antimicrobial peptide libraries.** The bioactive regions of enterocin AS-48 and AS-48 bacteriocin homologs were used as a scaffold for rationally designed antimicrobial peptide libraries as previously described (see Table S1 in the supplemental material) (50). Briefly, the Basic Local Alignment Search Tool (BLAST) was used to identify previously uncharacterized AS-48 bacteriocin homologs produced by *Bacillus safensis*, *Clostridium sordellii*, *Paenibacillus larvae*, and *Bacillus xiamenensis*. These minimal regions, along with the truncated bioactive region of enterocin AS-48, were then used as a scaffold for library design. Specific rational design strategies were employed for overall library design to optimize antimicrobial properties. First, aliphatic and nonpolar short-chain amino acids were replaced with either lysine or tryptophan, hypothesized to increase peptide electrostatic affinity and penetration into the phospholipid bilayer, respectively (69–71). Next, to increase the amphipathic nature of the peptide, amino acids were inverted within the helical wheel to aggregate hydrophobic amino acids. The clustering of hydrophobic amino acids on the peptide surface may facilitate insertion into the hydrophobic membrane core (72–74). In total, 95 synthetic 25-mer peptide variants of each scaffold peptide, designated syn-enterocin, syn-safencin, syn-sordellicin, syn-larvacin, and syn-xiamencin, were designed based on these defined biophysical parameters and commercially synthesized by GenScript (Piscataway, NJ). Synthesis was confirmed to be >95% pure and was verified by high-performance liquid chromatography–mass spectrometry (HPLC-MS) prior to use (GenScript). Lyophilized peptides were suspended in Nanopure water, diluted to a final stock concentration of 1.28 mM, and stored at −20°C.

**Parasite culture.** *Leishmania donovani* transgenic strain 1S2D (MHOM/SD/62/1S-CL2d) clone LdB constitutively expressing mCherry51 was cultured in M199 supplemented with 10% fetal calf serum (FCS) at 27°C at pH 7.4. Axenic amastigotes were differentiated as described previously (69).

**Antileishmanial peptide library screen and IC$_{50}$ quantification.** Axenic amastigotes were differentiated as described previously (69); briefly, transgenic *L. donovani* promastigotes expressing mCherry were differentiated by pH and temperature shifts. Amastigotes were seeded into 96-well plates at 5 × 10$^6$ cells/mL and incubated for 48 h in the presence of decreasing drug concentrations, starting at 20 $\mu$M, along with the appropriate solvent controls. Fluorescence was monitored (587-nm excitation/610-nm emission) over 48 h using a FlexStation 3 benchtop multimode reader (Molecular Devices). At 50 $\mu$M miltefosine, 99% parasite inhibition is observed. Thus, changes in mCherry fluorescence were normalized to the 50 $\mu$M miltefosine control in order to quantify parasite inhibition. Each assay was performed in triplicate. The IC$_{50}$ values were calculated by nonlinear regression analysis using GraphPad Prism 9.0 for Windows.

**Evaluation of cytotoxicity against THP-1 macrophages.** THP-1 (human acute monocytic leukemia-derived) host macrophages were cultured in RPMI 1640 supplemented with 10% FCS and a penicillin-streptomycin antibiotic solution (10,000 U/mL penicillin, 10,000 $\mu$g/mL streptomycin) at 37°C with 5% CO$_2$. THP-1 cells were incubated with 0.25 $\mu$M phorbol 12-myristate 7-acetate (PMA) for 48 h to differentiate into mature macrophages. Differentiated macrophages were seeded into 96-well plates at 5 × 10$^6$ cells/mL and incubated for 48 h in the presence of 20 $\mu$M antimicrobial peptides along with the appropriate solvent controls. Cell viability assays were conducted using CellTiter-Blue (Promega), based on resazurin reduction. After the 48-h incubation period, CellTiter-Blue reagent was added to cells for 4 h at 37°C, and fluorescence was measured (555-nm excitation/580-nm emission) using a FlexStation 3 benchtop multimode reader (Molecular Devices). Each assay was performed in triplicate. The CC$_{50}$ values were calculated by nonlinear regression analysis using GraphPad Prism 9.0 for Windows.

**Intracellular leishmanicidal activity analysis.** Transgenic *L. donovani* promastigotes constitutively expressing mCherry and THP-1 macrophages were cultured as described above. Metacyclic promastigotes were isolated using a Ficoll-400 density gradient as previously described (70). Differentiated macrophages were exposed to parasites for 4 h at a multiplicity of infection (MOI) of 10 parasites:1 macrophage. After 4 h, the macrophages were washed thoroughly with 1× phosphate-buffered saline (PBS) (pH 7.4) and incubated overnight at 37°C with 5% CO$_2$. Macrophages were transferred to 96-well plates at 5 × 10$^6$ cells/mL and exposed to antimicrobial peptides at 20 $\mu$M. The mCherry fluorescence was monitored (587-nm excitation/610-nm emission) over 48 h using a FlexStation 3 benchtop multimode reader (Molecular Devices). Changes in mCherry fluorescence were normalized to the 50 $\mu$M miltefosine control to quantify parasite inhibition, as 50 $\mu$M miltefosine corresponds to 99% intracellular parasite inhibition. Each assay was performed in triplicate. The IC$_{50}$ values were calculated by nonlinear regression analysis using GraphPad Prism 9.0 for Windows.

**Computational analysis of secondary structure and stability.** Secondary structure was computationally predicted using TrRosetta and visualized in PyMOL (71, 72). Hydrophobic, positive, and negative regions were colored in orange, blue, and red, respectively. To investigate the hydrophobic dipole moment ($\mu H$), which characterizes the amphipathic nature of the peptides, along with the mean hydrophobicity ($H$) and net charge ($z$), and to visualize the predicted helical wheel, HeliQuest was used (73).

To analyze stability, the generated .PDB files from TrRosetta were inputted into SCooP, a bioinformatic tool used to predict protein stability curves (74). The melting temperature (degrees Celsius) was used to predict thermostability, while the quantified change in the folding free energy ($\Delta G$) was used to determine thermodynamic stability.

## SUPPLEMENTAL MATERIAL

Supplemental material is available online only.
**SUPPLEMENTAL FILE 1**, PDF file, 1.5 MB.

## ACKNOWLEDGMENTS

We acknowledge the University of Notre Dame for its support.

This work was supported by the Young Innovator Award from the National Institutes of Health awarded to S.W.L. (NIH-1DP2OD008468-01). H.N.C. was supported in part by Indiana Clinical and Translational Sciences Institute-funded award number UL1TR002529 from the National Institutes of Health, National Center for Advancing Translational Sciences, Clinical and Translational Sciences Awards Program, and the Chemistry-Biochemistry-Biology Interface (CBBI) Program at the University of Notre Dame, supported by training grant T32GM075762 from the National Institute of General Medical Sciences.

We have no conflicts of interest to declare.

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
