## [Reviewer comments · Microbiology Spectrum]

Microbiology Spectrum

Rationally Designed Minimal Bioactive Domains of AS-48 Bacteriocin Homologs Possess Potent Anti-Leishmanial Properties

Hannah Corman, Jessica Ross, Francisco Fields, Douglas Shoue, Mary Ann McDowell, and Shaun Lee

Corresponding Author(s): Shaun Lee, University of Notre Dame

Review Timeline:

Submission Date:	July 11, 2022
Editorial Decision:	July 29, 2022
Revision Received:	September 8, 2022
Accepted:	October 4, 2022

Editor: Tiffany Weinkopff

Reviewer(s): Disclosure of reviewer identity is with reference to reviewer comments included in decision letter(s). The following individuals involved in review of your submission have agreed to reveal their identity: Greg Naumiec (Reviewer #2)

Transaction Report:

DOI: <https://doi.org/10.1128/spectrum.02658-22>

July 29, 2022

Dr. Shaun W. Lee
University of Notre Dame
Department of Biological Sciences
Notre Dame, IN 46556

Re: Spectrum02658-22 (Rationally Designed Minimal Bioactive Domains of AS-48 Bacteriocin Homologs Possess Potent Anti-Leishmanial Properties)

Dear Dr. Shaun W. Lee:

Link Not Available

Sincerely,

Tiffany Weinkopff

Journals Department
Reviewer comments:

Reviewer #1 (Comments for the Author):

The authors assayed successfully an extremely very well populated peptide library from several bacteriocins first, on *L.donovani* axenic amastigotes as a first selection step, to be tested later on intracellular parasites. From that 10 peptides were defined with excellent properties as leishmanicidal candidates. This result deserves publication once several issues will be addressed and amended, as they are incorrectly mentioned. See the list below :

1. Statistics must be provided, at least for the IC50s of intracellular amastigotes.

2. For the sake of comparison, the leishmanicidal activity of native bacteriocins should be tested and included in the same experimental system, to know how much the activity is improved by the approach used in this work.

3. According to the workflow scheme (fig 1) the number of peptides that fulfill the criteria to be tested on intracellular amastigote (SI >20, IC50 < 5 μ M) more than 40 peptides included within the Supplementary Tables 2-6 accomplished these criteria. Peptide availability was just the exclusive criterion to select 10 peptides?. Please comment on that, as many good, even better candidates than those tested, may have been lost.

4. As the best peptide from this work did not come from AS-48 but from safecin, I would suggest a change in the title of the work to be more inclusive and closer to the results.

ADDITIONAL ISSUES:

Line 58.- "Pentavalent antimonials as first-line drugs". The clinical use of Pentavalent antimonials is under progressive decline, being replaced by the liposomal form of amphotericin and miltefosine. Consequently, they may hardly be considered as first-line drugs.

Line 68.- The order of the concepts in this paragraph is confusing and may mislead the potential reader by not following a logical order. First, membrane active peptides as antiparasitic peptides was illustrated with one reference for each protozoan, then histatin 5 as intracellular peptide is mentioned, followed by bombinins with two references. Why the authors stressed the importance of bombinins respect to the subject of the current work?. If bombinins were membrane-active peptides should be included previously, where membrane-active antiprotozoal peptides were mentioned but if they have an intracellular target then these references, if essential, should be gathered with histatin5.

Line 107.- The efforts to develop synthetic circular variants of AS-48 (Rohrbacher 2017 Chem.Sci 8, 4051), an important issue to engineer improved versions of AS-48 were not mentioned. The same happens with the unsuccessful attempts to obtain linear forms of AS-48 by proteolysis (Montalban-Lopez 2008 FEBS Lett. 582, 3237, even when this failed attempt highlights the success obtained in the current work.

Line 117.- " and also exhibited IC50 values against axenic amastigotes of approximately 10.2 to 19.5 μ M". After checking the work mentioned, these values cannot be dubbed as "approximate". In fact, they are the values of IC50 and IC90 of native AS-48 on *Leishmania pifanoi* axenic amastigotes. In this tune, taking into account the number of replicates (three), to express IC50s with three decimal places is obviously meaningless from a microbiological experimentation. Otherwise, confidence intervals or standard deviation must be included.

Line 140.- "Rational design of AS-48 AMPs libraries" The process leading to the definition of these regions should be detailed, not only by the inclusion of one reference. This issue is crucial to understand the rationale followed in this approach, thus the fundamental of the process and the decision milestones should be described. The same happens with the "Bioinformatic tools" used for this purpose.

Line 157.- I wonder whether the poor reliability of amphipathicity as descriptor for leishmanicidal activity (line 278) may come from a feasible peptide aggregation, due to the mode of stock preparation, with a peptide concentration of 1.2 mM in pure water. Please, comment.

Line 173 Just as a comment and for future work. Miltefosine at 50 μ M is a quite an excessive concentration to kill intracellular amastigotes, in fact it may lead to distorted results by a detergent effect on macrophages and even amastigotes. Concentrations lower than 10 μ M are enough to provide a basal value for comparison.

Line 245.- IC50s below micromolar concentrations are hardly found for membrane -active AMPs as they they act through a stoichiometric perturbation of the phospholipid bilayer and must cover a substantial percentage of the whole surface. In this regard, if the peptides were aggregated in solution, as aforementioned, it may facilitate a high local concentration after insertion of the peptide aggregate at the membrane, and may account for active nanomolar concentrations, otherwise an intracellular target should be suspected as well. Do the authors may comment on that?. As a clue, nanomolar concentrations of AS-48 were described as active against *Trypanosoma brucei* (ref 32)

Line 250 "which exhibited MIC values below the previously characterized enterocin AS-48(ref 48)". In this reference, IC50s and IC90s were used but MIC were no mentioned at all. Please correct.

Line 342.-The role of physical parameters to account for the leishmanicidal activity of the 10 peptides included at Table 2 is difficult to follow. New plots accounting for the variation of the different physical parameters respect to IC50 at the Supplementary material will be quite illustrative.

Line 378.- "Upon electrostatic interactions peptides may polymerize (65)". Polymerization accounts for the formation of covalent

bond; aggregation is a more accurate term.

Line 379 The difference in the outer membrane lipid composition.- Outer membrane is mostly referred for Gram negative bacteria, cell membrane must be used, and more specific the outer hemilayer of the plasma or cell membrane.

Line 382.- As rightly mentioned lipophosphoglycan(LPG) is an abundant component of the plasma membrane of Leishmania, but only for promastigotes, not for amastigotes, the parasite stage used in this work, were it is replaced by glycosylphosphatidyl inositol lipids, much less cationic. Only few hundred copies of LPG were expressed at the amastigote stage, thus the argumentation on this issue must be eliminated or modified accordingly.

Line 383.- The mechanism of intracellular peptide accumulation by endocytosis is feasible, although not by fluid phase pinocytosis, once the authors discarded the interaction of AMP with the phospholipids of the cell membrane, but promoted by a preferential interaction of the cationic AMPs with the anionic syndecans or heparan sulfate components of the extracellular matrix of macrophages. Furthermore, the authors skip a discussion on the liability of natural peptides, especially those rich in basic residues to degradation by endosomal or lysosomal degradation. Please, comment this issue.

Line 388.- Enterocin AS-48 is not mentioned in the reference included(20). Thus to state that this reference supports the modification of membrane fluidity of protozoal membranes by AS-48 is not correct.

Reviewer #2 (Comments for the Author):

Just a few comments and suggestions...

1) Figures 1 and 3 need to be higher resolution images. When zooming in on them, they become blurry.

2) Be sure to have a space between numbers and units. This appears several times throughout the manuscript. Lines 159, 172, 173, 175, 182, 188, 189, 200, 202, 230, 295, 296, the IC50 units in figure 1.

3) When referring to the number of peptides variants, any number below 10 should be spelled out, not numerical. This occurs through the manuscript. Lines 236, 242, 252, 253, 256, 335, 338, 346, 347, 355

4) Materials and Methods section - line 143, define the bioinformatic tools that were used. Line 157, define the type of MS used (i.e. MALDI-HRMS). Line 175, you have two periods in a row. Line 182, CO₂ should have the 2 as a subscript. Line 198, define the concentration and pH of the PBS buffer.

Staff Comments:

Preparing Revision Guidelines

Please return the manuscript within 60 days; if you cannot complete the modification within this time period, please contact me. If you do not wish to modify the manuscript and prefer to submit it to another journal, please notify me of your decision immediately so that the manuscript may be formally withdrawn from consideration by Microbiology Spectrum.

Reviewer comments:

Reviewer #1 (Comments for the Author):

We want to thank Reviewer #1 for their comments about the major findings of our paper. Your interest in our work, as well as your thoughtful and detailed, constructive feedback to improve our manuscript is greatly appreciated. Your comments have been thoroughly addressed and incorporated into our revised manuscript. Specific comments regarding each point are noted below.

1. Statistics must be provided, at least for the IC₅₀s of intracellular amastigotes.

We have now added the standard deviation (SD) values for all calculated IC₅₀ values were added to the main-text figures and supplementary figures.

2. For the sake of comparison, the leishmanicidal activity of native bacteriocins should be tested and included in the same experimental system, to know how much the activity is improved by the approach used in this work.

We agree with the reviewer that comparing peptide bioactivity to the native bacteriocin is important to highlight the importance of our work. Although the bacteriocin Enterocin AS-48 has been purified and is well characterized, other Enterocin AS-48 homologues, such as AS-48 Safencin (the native bacteriocin for our studies) have not been purified in the same way. Much of the difficulty in purifying the native bacteriocin in large and pure yield come from the challenges in robust expression in native bacterial species, coupled with complexities in purification methods, especially for circular native bacteriocins such as those of the AS-48 family. Therefore, we chose to concentrate on the reduced synthetic peptides for our screening strategy, as they can be made in yield with high purity. Our previous studies showed that the bioactive region of Enterocin AS-48 can be reduced to a specific region of the natural bacteriocin, and therefore, is the basis for the library that we tested in this paper. Previous studies by other groups also established IC₅₀ values for the native bacteriocin (Enterocin AS-48) against *Leishmania* spp. and therefore, rather than repeat this study with Safencin AS-48 we chose to highlight the novel finding of showing that truncated peptides of AS-48 may serve as a scaffold to design peptides with increased bioactivity against parasitic organisms.

3. According to the workflow scheme (fig 1) the number of peptides that fulfill the criteria to be tested on intracellular amastigote (SI >20, IC₅₀<5 μM) more than 40 peptides included within the Supplementary Tables 2-6 accomplished these criteria. Peptide availability was just the exclusive criterion to select 10 peptides? Please comment on that, as many good, even better candidates than those tested, may have been lost.

There were multiple criteria used to select the 10 peptide candidates that were screened in the intracellular amastigote assay. A crucial criterion to consider was the toxicity to THP-1 macrophages and the resulting SI of those peptides. There were only

25 peptide candidates that had SI > 20, not more than 40 peptides as suggested: four Syn-enterocin candidates, three Syn-larvacin candidates, 10 Syn-safencin candidates, three Syn-sordellicin candidates, and five Syn-xiamencin candidates. Of those 25 peptide candidates, we chose 10 peptides that also exhibited low-micromolar to sub-micromolar IC₅₀ values against axenic amastigotes. We agree that we may have unintentionally missed some good or better candidates because of this selection process, but we will consider screening the remaining 15 candidates with SI > 20 against intracellular amastigotes in future studies, with additional libraries based on other bacteriocin templates. However, we wish to emphasize in this study, that the rational design-based reduced synthetic bacteriocin strategy presents an innovative strategy for discovering novel drug candidates against *Leishmania* spp.

4. As the best peptide from this work did not come from AS-48 but from safencin, I would suggest a change in the title of the work to be more inclusive and closer to the results.

Thank you for this suggestion, the canonical bacteriocin produced by *Enterococcus faecalis* is named “Enterocin AS-48,” thus we believe using “AS-48 bacteriocin homologs” refers to all scaffold peptides and is inclusive of safencin because as the reviewer correctly notes, the Safencin library was our most successful peptide library.

ADDITIONAL ISSUES:

Line 58.- "Pentavalent antimonials as first-line drugs". The clinical use of Pentavalent antimonials is under progressive decline, being replaced by the liposomal form of amphotericin and Miltefosine. Consequently, they may hardly be considered as first-line drugs.

This sentence has been re-written to remove the word “first-line.” We do believe in resource-limited places, where more expensive drugs like Miltefosine are not affordable to the patient population, pentavalent antimonials are still prescribed.

Line 68.- The order of the concepts in this paragraph is confusing and may mislead the potential reader by not following a logical order. First, membrane active peptides as antiparasitic peptides were illustrated with one reference for each protozoan, then histatin 5 as intracellular peptide is mentioned, followed by bombinins with two references. Why the authors stressed the importance of bombinins respect to the subject of the current work? If bombinins were membrane-active peptides should be included previously, where membrane-active antiprotozoal peptides were mentioned but if they have an intracellular target then these references, if essential, should be gathered with histatin5.

We appreciate the feedback for the structure of this paragraph. We recognize the order was confusing due to the omitted information that Histatin 5 is also membrane-active and causes damage to membrane and consequential depolarization of the membrane.

This has been added for clarity. This paragraph services to highlight all membrane-active AMPs with activity against *Leishmania spp.*, specifically.

Line 107.- The efforts to develop synthetic circular variants of AS-48 (Rohrbacher 2017 Chem.Sci 8, 4051), an important issue to engineer improved versions of AS-48 were not mentioned. The same happens with the unsuccessful attempts to obtained linear forms of AS-48 by proteolysis (Montalban-Lopez 2008 FEBS Lett. 582, 3237, even when this failed attempt highlights the success obtained in the current work.

The engineering efforts of AS-48 has now been expanded to add Rorhrbacher et al 2017 work and further explanation of the connection between our study and Montalban-Lopez research.

Line 117.- " and also exhibited IC50 values against axenic amastigotes of approximately 10.2 to 19.5 μM ". After checking the work mentioned, these values cannot be dubbed as "approximate". In fact, they are the values of IC50 and IC90 of native AS-48 on *Leishmania pifanoi* axenic amastigotes. In this tune, taking into account the number of replicates (three), to express IC50s with three decimal places is obviously meaningless from a microbiological experimentation. Otherwise, confidence intervals or standard deviation must be included.

Thank you for this correction. We have updated the text to emphasis that an IC50 value of $10.2 \pm 1.2 \mu\text{M}$ was observed against *L. pifanoi*.

Line 140.-"Rational design of AS-48 AMPs libraries" The process leading to the definition of these regions should be detailed, not only by the inclusion of one reference. This issue is crucial to understand the rationale followed in this approach, thus the fundamental of the process and the decision milestones should be described. The same happens with the "Bioinformatic tools" used for this purpose.

Thank you for this comment, we absolutely agree that the rationale behind the design should be emphasized. As such, the bioinformatic tool used to identify homologous regions in AS-48-like bacteriocins has been defined. We have also further added citations which highlight the rationale behind the biophysical rational design process of reduced bacteriocin design, a central feature of our work.

Line 157 .- I wonder whether the poor reliability of amphipathicity as descriptor for leishmanicidal activity (line 278) may come from a feasible peptide aggregation, due to the mode of stock preparation, with a peptide concentration of 1.2 mM in pure water. Please, comment.

We appreciate the reviewer's comment and insightful thought on this. We do believe that the poor reliability of amphipathicity as a descriptor for anti leishmanicidal activity is not due to aggregation because of the stock concentration. Our peptides are confirmed

to be soluble in water at this concentration. Previously, we have tested these libraries against bacteria, and we were able to conclude that increases to amphipathicity did play a role in optimization in targeting bacterial membranes. We believe that the range of amphipathicity that we included may not serve as a highly important factor in anti-leishmanial activity, most likely due to the nature of the membrane composition and charge distribution of the Leishmania membrane surface, in contrast to the bacterial membrane. The reviewer makes an important point however, which is how the parameters for peptide design can be amended or optimized to address specifically the design of these libraries for anti-parasitic vs. antibacterial activity. We plan to pursue these questions in future studies.

Line 173 Just as a comment and for future work. Miltefosine at 50 μM is a quite an excessive concentration to kill intracellular amastigotes, in fact it may lead to distorted results by a detergent effect on macrophages and even amastigotes. Concentrations lower than 10 μM are enough to provide a basal value for comparison.

Thank you for the feedback. When continuing this work in the future, we will certainly consider using the minimum concentration of Miltefosine to see inhibition. Previous work in establishing our fluorometric assays against promastigotes and axenic amastigotes revealed that 99% of the mCherry signal was inhibited at 50 μM Miltefosine; we then continued using this concentration for normalization to ensure consistency among studies.

Line 245.- IC50s below micromolar concentrations are hardly found for membrane-active AMPs as they act through a stoichiometric perturbation of the phospholipid bilayer and must cover a substantial percentage of the whole surface. In this regard, if the peptides were aggregated in solution, as aforementioned, it may facilitate a high local concentration after insertion of the peptide aggregate at the membrane, and may account for active nanomolar concentrations, otherwise an intracellular target should be suspected as well. Do the authors may comment on that? As a clue, nanomolar concentrations of AS-48 were described as active against Trypanosoma brucei (ref 32)

Thank you for this important point, and we agree with the reviewer that potential multimerization of our peptides on the surface of membranes would lead to facilitation of a high local concentration of peptides and the assembly of larger, more substantial holes in the membrane that would account for activity at nanomolar concentrations. In a previous report, we indeed showed through patch clamp studies of model bacterial membranes, that large pores are formed for some peptide candidates on the bacterial surface, that substantially enhance bacterial activity (Fields et al. ACS Pharm Transl, 2020). Although we do not see any appreciable aggregation of the peptides tested in this study in solution, it is intriguing to hypothesize that binding interactions and lateral membrane movement might facilitate the multimerization of the peptides on the surface of the parasitic membrane. Although this is beyond the scope of our current study, this is an important point that will give key insights into the mechanisms by which our

peptides act on the membrane of parasites, given their low IC values. We plan to investigate this in future studies.

The text has also now been updated to include information about potential intracellular targets as suggested.

Line 250 "which exhibited MIC values below the previously characterized enterocin AS-48(ref 48)". In this reference, IC50s and IC90s were used but MIC were no mentioned at all. Please correct.

Thank you for pointing this error, the MIC has now been replaced with the IC50 value.

Line 342.-The role of physical parameters to account for the leishmanicidal activity of the 10 peptides included at Table 2 is difficult to follow. New plots accounting for the variation of the different physical parameters respect to IC50 at the Supplementary material will be quite illustrative.

We understand that the physical parameters may be difficult to follow without doing a side-by-side comparative with the IC50 values; however, we felt that including the SI index information in Figure 2 and providing structural illustrations in Figure 3 allowed us to separate the information in a meaningful, and simple manner, such that the peptide candidates with the best activity can be identified easily in Figure 3. While table 2 also summarizes the key parameters in the peptide candidates, there is no strong trend between these values, the structures in Figure 2, and the SI and IC50 values and therefore we chose to highlight each dataset as individual figures and tables for clarity.

Line 378.- "Upon electrostatic interactions peptides may polymerize (65)". Polymerization accounts for the formation of covalent bond; aggregation is a more accurate term.

We agree that aggregation is a more accurate term, and the text has been updated to reflect this.

Line 379 The difference in the outer membrane lipid composition.- Outer membrane is mostly referred for Gram negative bacteria, cell membrane must be used, and more specific the outer hemilayer of the plasma or cell membrane.

Thank you for this comment. Outer membrane has been replaced with cell membrane.

Line 382.- As rightly mentioned lipophosphoglycan(LPG) is an abundant component of the plasma membrane of Leishmania, but only for promastigotes, not for amastigotes, the parasite stage used in this work, where it is replaced by glycosylphosphatidyl inositol lipids, much less cationic. Only few hundred copies of LPG were expressed at the amastigote stage, thus the argumentation on this issue must be eliminated or modified accordingly.

Thank you for the feedback. We agree that the limited amount of LPG on amastigotes likely means that the density of peptides near the plasma membrane would be lower than what is needed to create large, systemic effects to the parasite. We have updated the text to include this information as well as to include a statement about possible intracellular targets that were not elucidated in the current study.

Line 383.- The mechanism of intracellular peptide accumulation by endocytosis is feasible, although not by fluid phase pinocytosis, once the authors discarded the interaction of AMP with the phospholipids of the cell membrane, but promoted by a preferential interaction of the cationic AMPs with the anionic syndecans or heparan sulfate components of the extracellular matrix of macrophages. Furthermore, the authors skip a discussion on the liability of natural peptides, especially those rich in basic residues to degradation by endosomal or lysosomal degradation. Please, comment this issue.

We thank the reviewer for this insightful comment. We have now included a discussion of the potential macrophage interactions into the text. We did not comment on the liability of natural peptides, since our study focuses on smaller peptide variants being truncated, altered versions of natural peptides designed to increase net charge, hydrophobicity, and the hydrophobic moment. Our previous studies and others who have designed small, reduced peptides based on larger natural bacteriocins have implemented these strategies such that stability and resistance to degradation and proteolysis can be improved by these shorter peptide variants. We have not performed any *in vivo* studies on our peptide candidates as they were beyond the scope of this study; however, the reviewer makes a key point that our future studies using *in vivo* approaches will need to address issues of bioavailability, stability, and in particular how the peptides can be stabilized while inside a maturing phagolysosome. Currently, we are employing strategies to conjugate the peptides to nanoparticles to facilitate their stability in intracellular vesicles to enhance potential targeting to intracellular *Leishmania*. It is also interesting to note that *Leishmania* parasites have evolved several mechanisms to prevent the maturation of phagolysosomes within macrophages, including LPG interaction with membranes to impair recruitment of necessary phagosomal machinery like GTPases, and this may somehow inadvertently improve the ability of our peptides to remain stable, and target the intracellular parasite.

Line 388.- Enterocin AS-48 is not mentioned in the reference included(20). Thus, to state that this reference supports the modification of membrane fluidity of protozoal membranes by AS-48 is not correct.

This has been removed and replaced with a relevant citation.

Reviewer #2 (Comments for the Author):

We thank Reviewer #2 for their comments and edits to improve our manuscript. Your comments have been incorporated into our revised manuscript. Specific comments regarding each point are noted below.

1) Figures 1 and 3 need to be higher resolution images. When zooming in on them, they become blurry.

Thank you for pointing out that the resolution needs to be improved. The figures have been adjusted to have a higher resolution have been changed to higher resolution TIFF files, which should be clearer when zooming in.

2) Be sure to have a space between numbers and units. This appears several times throughout the manuscript. Lines 159, 172, 173, 175, 182, 188, 189, 200, 202, 230, 295, 296, the IC50 units in figure 1.

Thank you for noting this, a space has been added between number and unit throughout the paper now.

3) When referring to the number of peptides variants, any number below 10 should be spelled out, not numerical. This occurs through the manuscript. Lines 236, 242, 252, 253, 256, 335, 338, 346, 347, 355

Thank you for this comment, we have updated the manuscript to reflect this comment.

4) Materials and Methods section - line 143, define the bioinformatic tools that were used. Line 157, define the type of MS used (i.e., MALDI-HRMS). Line 175, you have two periods in a row. Line 182, CO₂ should have the 2 as a subscript. Line 198, define the concentration and pH of the PBS buffer.

Thank you for noting this correction. The text has now been updated to reflect the bioinformatics tool used (BLAST). The verification process of the peptides was confirmed using HPLC-MS, the method has been updated in text. The extra period has been deleted. The concentration and pH of the PBS buffer was updated.

October 4, 2022

Dr. Shaun W. Lee
University of Notre Dame
Department of Biological Sciences
Notre Dame, IN 46556

Re: Spectrum02658-22R1 (Rationally Designed Minimal Bioactive Domains of AS-48 Bacteriocin Homologs Possess Potent Anti-Leishmanial Properties)

Dear Dr. Shaun W. Lee:

Your manuscript has been accepted, and I am forwarding it to the ASM Journals Department for publication. You will be notified when your proofs are ready to be viewed.

Sincerely,

Tiffany Weinkopff
Editor, Microbiology Spectrum
